# Beyond Predictions in Neural ODEs: Identification and Interventions

## Abstract

Spurred by tremendous success in pattern matching and prediction tasks, researchers increasingly resort to machine learning to aid original scientific discovery. Given large amounts of observational data about a system, can we uncover the rules that govern its evolution? Solving this task holds the great promise of fully understanding the causal interactions and being able to make reliable predictions about the system's behavior under interventions. We take a step towards such system identification for time-series data generated from systems of ordinary differential equations (ODEs) using flexible neural ODEs. Neural ODEs have proven successful in learning dynamical systems in terms of recovering observed trajectories. However, their efficacy in learning ground truth dynamics and making predictions under unseen interventions are still underexplored. We develop a simple regularization scheme for neural ODEs that helps in recovering the dynamics and causal structure from time-series data. Our results on a variety of (non)-linear first and second order systems as well as real data validate our method. We conclude by showing that we can also make accurate predictions under interventions on variables or the system itself.

## 1    Introduction

Many research areas increasingly embrace data-driven machine learning techniques not only for prediction, but also with the hope of leveraging data for original scientific discoveries. We may formulate the core task of an "automated scientist" as follows: *Given observational data about a system, identify the underlying rules governing it!* A key part of this quest is to determine how variables depend on each other. Putting this question at its center, causality is a natural candidate to surface scientific insights from data.

Great attention has been given to "static" settings, where each observable under consideration is a random variable whose distribution is given as a deterministic function of other variables—its causal parents. The structure of "which variables listen to what other variables" is typically encoded as a directed acyclic graph (DAG), giving rise to graphical structural causal models (SCM) (Pearl, 2009). SCMs have been successfully deployed both for inferring causal structure as well as estimating the strength of causal effects. However, in numerous scientific fields, we are interested in systems that jointly evolve over time with dynamics governed by differential equations, which cannot be easily captured in a standard SCM. In such interacting systems, the instantaneous derivative of a variable is a function of other variables (and their derivatives). We can thus interpret the variables (or derivatives) that enter this function as causal parents (Mooij et al., 2013). Unlike static SCMs, this accommodates cyclic dependencies and temporal co-evolution (Bongers et al., 2016).

The hope is that machine learning may sometimes be able to deduce true laws of nature purely from observational data, promising reliable predictions not only within the observed setting, but also under interventions.[1] In this work we take a step towards system identification and causal structure inference from time-series data of a *fully observed* system that is assumed to *jointly evolve according to an ODE*. We use scalable and flexible neural ODE estimators (Chen et al., 2018) that allow for *a-priori unknown non-linear*

---

[1] An automated scientist would undoubtedly be more powerful if it can interact with the system and perform experiments. By focusing on the purely observational setting, we avoid an ad-hoc specification of which experiments can be conducted and adhere to current settings, where algorithms are not granted direct access to real-world interventions.

*interactions.* We start with noise-free observations in continuous time, but also provide results for additive observation noise, irregularly sampled time points, and real gene expression data. Generally, recovering the ODE from a single observed solutions is an ill-posed problem. However, existing theory suggests (informally) that for certain classes of ODEs "most systems" will be identifiable. Motivated by these findings, we make the following contributions:

- We discuss potential regularizers to enforce sparsity in the number of causal interactions such that variables depend on few other variables, a common assumption in modern causal modeling Schölkopf et al. (2021).

- We develop a causal structure inference technique from time-series data called *causal neural ODE* (**C-NODE**) combining flexible neural ODE estimators with suitable regularization techniques. C-NODE works for non-linear ODEs with cyclic causal structure and identifies the full ODE in certain cases. Our results suggest that the proposed regularizer improves identifiability of the governing dynamics.

- We demonstrate the efficacy of C-NODE on a variety of low-dimensional (non-)linear first and second order systems, where it also makes accurate predictions under *unseen interventions*. On simulated autonomous, linear, homogeneous systems we show that C-NODE scales to tens of variables.

- Finally, C-NODE yields promising results for gene regulatory network inference on real, noisy, irregularly sampled single-cell RNA-seq data.

**Related work.** There is a large body of work on the discovery of causal DAGs within the static SCM framework (Heinze-Deml et al., 2018; Glymour et al., 2019; Vowels et al., 2021). One key idea for causal discovery on time-series data is based on *Granger causality*, where we attempt to forecast one time-series based on past values of others (Granger, 1988). We review the basic ideas and contemporary methods in Section 2.3. Typically, these methods also return an acyclic directed interaction model, though feedback of the forms $X_t \to X_{t+1}$ or $X_t \to Y_{t+1}$ and $Y_t \to X_{t+1}$ is allowed. Inferring Granger causality often relies on conditional independence tests (Malinsky & Spirtes, 2018) or score-based methods (Pamfil et al., 2020). Certain extensions of SCMs to cyclic dependence structures that retain large parts of the causal interpretation (Bongers et al., 2016) also allow for causal discovery of cyclic models (Lacerda et al., 2012; Hyttinen et al., 2012; Mooij et al., 2011). The framing of our work differs from the above in that they cannot model evolutions of instantaneously interacting systems and only aim at the causal structure instead of the system specifics.

Another line of research has explored connections between (asymptotic) equilibria of differential equations and (extended) SCMs that preserve behavior under interventions (Mooij et al., 2013; Bongers & Mooij, 2018; Rubenstein et al., 2018; Blom et al., 2020). Pfister et al. (2019) focus on recovering the causal dependence of a single target variable in a system of differential equations by leveraging data from multiple heterogeneous environments. Following earlier work (Dondelinger et al., 2013; Raue et al., 2015; Benson, 1979; Ballnus, 2019; Champion et al., 2019), they consider mass-action kinetics only taking into account linear combinations of up to degree one interactions of the target variable's parents. They enforce sparsity by only allowing a fixed number of such terms to be non-zero. More broadly, parameter identifiability and estimation has been thoroughly investigated for discrete dynamical systems (McGoff et al., 2015), when the entire solution space is available (Grewal & Glover, 1976), or time-lag is required with no instantaneous interactions (Runge, 2018; Ye et al., 2015). Building on Takens (1981), the "convergent cross mapping method" (Sugihara et al., 2012) overcomes separability assumptions of Granger causality and has successfully been extended to identify causal structure (and sometimes systems) for chaotic, strongly non-linear, or time-lagged systems among others (Ye et al., 2015; Runge et al., 2019; De Brouwer et al., 2020). Current methods for ODE parameter estimation (with known parametric form) often deal with structural (Cobelli & Distefano, 1980) and practical (e.g., partial observability (Raue et al., 2009)) non-identifiability via empirical uncertainty analysis (Raue et al., 2015). For example, Sindy (Brunton et al., 2016), a popular sparse regression method for identification of nonlinear systems, explicitly assumes sparsity in a set of candidate basis functions. This poses a limitation on the scalability of the method as the dictionary size grows combinatorially in the number of variables that are allowed to interact and nested non-linearities have to be added explicitly to the dictionary. Hence, Brunton et al. (2016) acknowledge that Sindy fails to recover the dynamics already for a system of seven variables and that allowing for "a broader function search space is an important area of current and future work".

In contrast to the works above, we assume neither a semantically meaningful pre-specified parametric form of the ODEs with a small set of parameters, nor the existence of equilibria. We consider fully observed, non-delay

ODEs (no time-lag, only instantaneous interactions) with observations from a single environment. Our focus is on scalability to many variables with unknown non-linearities and leverage neural networks as flexible, yet efficiently learnable, function approximators.[2] Finally, we aim at fully identifying the ODE system, not only the causal structure, to be able to make predictions under interventions with a special focus on the multivariate case, i.e., systems of ODEs with sparse dependency structures. Note that the causal structure is implied by the ODE system, hence ODE identification is strictly harder than causal structure identification in our setting. In most applications, we consider the causal structure as an informative byproduct of our attempt at ODE system identification.

Vorbach et al. (2021); Massaroli et al. (2021); Bellot et al. (2021) aim at interpreting the inner workings of NODEs. Our main difference to Vorbach et al. (2021); Massaroli et al. (2021) is that they do not go beyond predictive performance and disregard whether the true system has been learned. Vorbach et al. (2021); Bellot et al. (2021) do not discuss the behavior of the system under interventions.

This research has potential applications in diverse fields including biology (Pfister et al., 2019) (e.g., gene-regulatory network inference (Matsumoto et al., 2017; Qiu et al., 2020)), robotics (Murray et al., 1994; Kipf et al., 2018), and economics (Zhang, 2005).

## 2 Setup and Background

Assume we observe the temporal evolution of $n$ real-valued variables $X^\star : [a, b] \to \mathbb{R}^n$ on a continuous time interval $a < b$, such that $X^\star$ solves the system of ODEs[3]

$$\dot{X} = f^\star(X, t), \text{ for } f^\star \in \mathcal{F} \text{ where} \tag{1}$$
$$\mathcal{F} := \{f : \mathbb{R}^n \times [a, b] \to \mathbb{R}^n \mid f \text{ uniformly Lipschitz-continuous in } X \text{ and continuous in } t\}.$$

That is, we observe a single solution trajectory of some ODE (determined by) $f^\star \in \mathcal{F}$.

Our **main goal** is the following identification task: Given $X^\star$, identify $f^\star \in \mathcal{F}$. $\tag{2}$

This is the inverse problem of "solving an ODE", for which the celebrated Picard-Lindelöf theorem guarantees the existence and uniqueness of a solution of the *initial value problem* (IVP) $\dot{X} = f(X, t), X(a) = x_0$ for all $f \in \mathcal{F}$ and $x_0 \in \mathbb{R}^n$ on an interval $t \in (a - \epsilon, a + \epsilon)$ for some $\epsilon > 0$. We remark that higher-order ODEs, in particular second-order ODEs $\ddot{X} = f(X, \dot{X}, t)$, can be reduced to first-order systems via $U := (X, \dot{X}) \in \mathbb{R}^{2n}$, $\dot{U} = (\dot{X}, \ddot{X}) = (U_2, f(U, t))$.[4] Hence, it suffices to continue our analysis for first-order systems.

### 2.1 Causal Interpretation

In SCMs causal relationships are typically described by directed parent-child relationships in a DAG, where the causes (parents) of a variable $X_i$ are denoted by $pa(X_i) \subset X$. For ODEs an analogous relationship can be described by which variables "enter into $f_i$". Formally, we define the **causal parents of $X_i$ in system $f$**, denoted by $pa_f(X_i)$, as follows: $X_j \in pa_f(X_i)$ if and only if there exist $x_1, \ldots, x_{j-1}, x_{j+1}, \ldots, x_n \in \mathbb{R}$ such that $f_i(x_1, \ldots, x_{j-1}, \bullet, x_{j+1}, \ldots, x_n) : \mathbb{R} \to \mathbb{R}$ is not constant. This notion analogously extends to second and higher order equations by defining $pa_f(X_i)$ as the variables $X_j$ for which any (higher order) derivative of $X_j$ enters $f_i$. Thereby, identifying $f^\star$ in eq. (2) also yields the causal structure—it is an immediate yet informative byproduct.

One of the key advantages of causal models compared to merely predictive ones is that they enable us to make predictions about hypothetical interventions not in the training data. In the ODE setting, different types of interventions can be conceived of. We will focus on the following types of interventions.

---

[2]Local minima are always a concern in high-dimensional non-convex optimization (training), but orthogonal to our work.

[3]We use $X^\star$ for the real observed function and $X$ for a generic function. We refer to components $X_i$ as the observed variables of interest. Similarly, $f^\star$ is the ground truth system and $f$ a generic one. Lower case letters denote observations at a fixed time, e.g., $x_0 = X(t = 0)$.

[4]Second order systems require $X(a)$ and $\dot{X}(a)$ as initial values for a unique solution. In practice, when only $X^\star$ is observed, we assume that we can infer $\dot{X}^\star(a)$, either from forward finite differences or during NODE training, see Section 2.2. Any higher-order ODE can iteratively be reduced to a first order system.

- **Variable interventions:** For one or multiple $i \in \{1, \ldots, n\}$, we fix $X_i := c_i, f_i := 0$ and replace every occurrence of $X_i$ in the remaining $f_j$ with $c_i$ (for some constant(s) $c_i \in \mathbb{R}$). We interpret these interventions as externally clamping certain variables to a fixed value.
- **System interventions:** We replace one or multiple $f_i$ with $\tilde{f}_i$. Here we can further distinguish between *causality preserving* system interventions in which the causal parents remain unchanged, that is $pa_f(X_i) = pa_{\tilde{f}}(X_i)$ for all $i$, and others.

As an illustration, consider an ODE describing the positions of masses in a spring-mass system. A variable intervention could amount to externally keeping one mass at a fixed point in space. A system intervention could describe changing the stiffness of some of the springs. We will analyze variable interventions in (non-linear) ODEs and system interventions primarily in linear settings with interpretable system parameters like in the chemical reaction example.

## 2.2 (Neural) Ordinary Differential Equations

In neural ODEs (NODE) a machine learning model (often a neural network with parameters $\theta$) is used to learn the function $f_\theta \approx f$ from data (Chen et al., 2018). Starting from the initial observation $X^\star(a)$, an explicit iterative ODE solver is applied to predict $X^\star(t)$ for $t \in (a, b]$ using the current derivative estimates from $f_\theta$. The parameters $\theta$ are then updated via backpropagation on the mean squared error between predictions and observations. We mostly build on an augmented variant called SONODE that also works for second order systems and estimates the initial values $\dot{X}^\star(a)$ in an end-to-end fashion (Norcliffe et al., 2020).

Recently, NODEs have been extended to irregularly-sampled timesteps (Rubanova et al., 2019), stochastic DEs (Li et al., 2020; Oganesyan et al., 2020), partial DEs (Sun et al., 2019), Bayesian NODEs (Dandekar et al., 2020), and delay ODEs (Zhu et al., 2021). While these extensions could benefit our method, we use vanilla SONODE to disentangle the performance of our method from tuning the underlying NODE method. As discussed extensively in the literature, NODEs can outperform traditional ODE parameter inference techniques in terms of reconstruction error, especially for non-linear $f$ (Chen et al., 2018; Dupont et al., 2019). A subsequent advantage over previous methods is that we need not pre-suppose a parameterization of $f$ in terms of a small set of semantically meaningful parameters.

Recently, a number of regularization techniques has been proposed for NODEs, where NODEs are viewed as infinite depth limits of residual neural networks (typically for classification) and there is no single true underlying dynamic law (Finlay et al., 2020; Kelly et al., 2020; Ghosh et al., 2020; Pal et al., 2021; Grathwohl et al., 2019). We emphasize that these existing techniques regularize the number of model evaluations to improve efficiency in learning one out of many possible dynamics yielding good performance on a downstream predictive task. They are unrelated to our regularizer, which aims at identifying a single true underlying dynamical system from time series data. Our regularizer targets neither the number of function evaluations, nor sparsity in the weights of the neural network directly (like pruning), but the number of causal dependencies. Finally, another alternative to train neural networks that depend on few inputs, regularizing input gradients (Ross et al., 2017b; Ross & Doshi-Velez, 2017; Ross et al., 2017a) does not scale to high-dimensional regression tasks.

## 2.3 Granger Causality

Granger causality is a classic method for causal discovery in time series data that primarily exploits the directionality of time (Granger, 1988). Informally, a time series $X_i$ *Granger causes* another time series $X_j$ if predicting $X_j$ becomes harder when excluding the values of $X_i$ from a universe of all time series. Assuming that $X$ is stationary, multivariate Granger causality analysis usually fits a vector autoregressive model

$$X(t) = \sum_{\tau=0}^{k} W^{(\tau)} X(t - \tau) + E(t), \tag{3}$$

where $E(t) \in \mathbb{R}^n$ is a Gaussian random vector and $k$ is a pre-selected maximum time lag. We seek to infer the $W^{(\tau)} \in \mathbb{R}^{n \times n}$ from $X(t)$. In this setting, we call $X_i$ a Granger cause of $X_j$ if $|W_{i,j}^{(\tau)}| > 0$ for some

$\tau \in \{0, \ldots, k\}$. We need to ensure that $W^{(0)}$ encodes an acyclic dependence structure to avoid circular dependencies at the current time. Pamfil et al. (2020) then estimate the parameters in eq. (3) via

$$\min_{W} \left\| X(t) - \sum_{\tau=0}^{k} W^{(\tau)} X(t - \tau) \right\|_2 + \xi \|W^{(0)}\|_{1,1} + \rho \|W^{\backslash 0}\|_{1,1} \,, \tag{4}$$

where $W = (W^{(\tau)})_{\tau=0}^{k}$, $W^{\backslash 0} = (W^{(\tau)})_{\tau=1}^{k}$, and $\|\cdot\|_{1,1}$ is the element-wise $\ell_1$ norm, which is used to encourage sparsity in the system. In addition, to ensure that the graph corresponding to $W^{(0)}$ interpreted as an adjacency matrix is acyclic, a smooth score encoding "DAG-ness" proposed by Zheng et al. (2018) is added with a separate regularization parameter. While extensions to nonlinear cases exist (Diks & Wolski, 2016), we primarily compare to `Dynotears` by Pamfil et al. (2020)—a choice motivated further in Appendix A.

## 3  Theoretical Considerations

First, we note that our main goal in eq. (2) is ill-posed. A solution exists by assumption, but it may not be unique.[5] We provide a simple example of two different autonomous, linear, homogeneous systems that have at least one solution in common in Appendix B (where we provide all proofs).[6] This means that the underlying system is *unidentifiable* from observational data.

Autonomous, linear, homogeneous systems are worthy a closer look:

$$\mathcal{F}_{\mathrm{lin}} := \{f(X) = AX \mid A \in \mathbb{R}^{n \times n}\} \subset \mathcal{F}. \tag{5}$$

First, they are common models for chemical reactions or oscillating physical systems. Second, unlike for larger classes of ODEs, identifiability is reasonably well understood. Within $\mathcal{F}_{\mathrm{lin}}$ we can use $A$ and $f$ interchangeably. For such systems, Stanhope et al. (2014) developed beautiful graphical criteria for the identifiability of the system $A$ given $X^\star$, namely that the trajectory $X^\star$ is not confined to a proper subspace of $\mathbb{R}^n$. Qiu et al. (2022) recently showed that an equivalent characterization is that $A$ has $n$ distinct eigenvalues, which implies that almost all $A \in \mathbb{R}^{n \times n}$ are uniquely identifiable from $X^\star$ for almost all initial values $x_0 \in \mathbb{R}^n$.[7] In this sense, all unidentifiable systems are non-generic and likely require some "fine-tuning" like our example in Appendix B, indicating that non-identifiability may not be a prevalent issue for the average case in practice. While these results largely extend to affine linear systems (Duan et al., 2020), little is known about identifiability in general non-linear systems (Miao et al., 2011). One may suspect non-identifiability to be a greater issue there, but highly non-linear or even chaotic systems are sometimes known to be identifiable (Takens, 1981; Sugihara et al., 2012; De Brouwer et al., 2020).

While these results are encouraging, we are particularly interested in "simple" interactions, which we capture by sparsity. That is, we assume that in natural systems each variable depends on only few other variables as causal parents (Schölkopf et al., 2021). Counting the number of parent-child relationships in a system $f$ as $\|f\|_{\mathrm{causal}} := \sum_{i=1}^{n} |pa_f(X_i)|$, we are thus interested only in possible ground truths $f^\star$ with small $\|f^\star\|_{\mathrm{causal}}$ (compared to $n^2$). For $\mathcal{F}_{\mathrm{lin}}$, this amounts to matrices $A \in \mathbb{R}^{n \times n}$ with at most $k$ non-zero entries. To the best of our knowledge, it remains an open problem whether among such sparse matrices still almost all of them are identifiable from $X^\star$ (for almost all initial conditions $x_0$). Since sparse matrices are more likely to have repeated eigenvalues, the existing theory for dense matrices does not carry over to our setting. Hence, there is a gap between predictive performance (reconstruction error of $X$) and *identifying the governing system*, which is required to make *predictions under interventions*. NODEs perform well in terms of predictive performance, but theoretically they may do so by learning the "false" system. This is a key motivation for our empirical analysis in this work. In the following, we develop a method to identify such sparse systems from a single

---

[5]This violates one of the three Hadamard properties for well-posed problems. We use the word 'solution' somewhat ambiguously and care must be taken not to confuse solutions to a given ODE or IVP (find $X$ given $f \in \mathcal{F}$) and a solution of our main goal (finding $f^\star \in \mathcal{F}$ given $X^\star$).

[6]*Autonomy* means that $f$ does not explicitly depend on time $f(X, t) = f(X)$. *Linear* systems are ones where $f$ is linear in $X$, i.e., $f = A(t)X + b(t)$. *Homogeneous* systems are linear systems in which $b(t) = 0$.

[7]Specifically, this holds with respect to the Lebesgue measure on $\mathbb{R}^{n \times n}$ and $\mathbb{R}^n$. The result still holds for probability measures from most common random matrix ensembles such as the Gaussian orthogonal, the Wishart, or the Ginibre ensembles.

observed solution trajectory via regularization and assess identifiability empirically not only in the linear, but also non-linear case.

We first formulate our **regularized goal**: Given $X^\star$, find $f \in \mathcal{F}$ such that $X^\star$ solves $\dot{X} = f(X, t)$ and $f \in \arg\min_{g \in \mathcal{F}} \|g\|$ for some measure of complexity $\|\cdot\| : \mathcal{F} \to \mathbb{R}_{\geq 0}$. Without knowing $f^\star$ a priori, $\|\cdot\|_{\text{causal}}$ is difficult to enforce as a complexity measure in practice. Instead, we approximate the requirement of "not being constant w.r.t. to an argument" in our definition of causal parents via the following regularizer

$$\|f\|_\epsilon := \sum_{i,j=1}^n \mathbf{1}\{\|\partial_j f_i\|_2 > \epsilon\} \quad \text{for some } \epsilon \geq 0, \tag{6}$$

where $\|\cdot\|_2$ is the $L^2$ norm on the Hilbert space of square-integrable real functions (with respect to the Lebesgue measure). For $A \in \mathcal{F}_{\text{lin}}$ this captures sparsity in the common sense $\|A\|_{\epsilon=0} = \sum_{i,j=1}^n \mathbf{1}\{A_{ij} \neq 0\} = \|A\|_{\text{causal}}$.[8] There are still two hurdles to implementing $\|f\|_\epsilon$ in practice. (a) For the full non-linear case $f \in \mathcal{F}$ the $L^2$-norms of partial derivatives are difficult to evaluate efficiently and accurately. (b) Even for $A \in \mathcal{F}_{\text{lin}}$, $\|A\|_\epsilon$ is not differentiable. For the linear case, non-differentiability of $\|A\|_\epsilon$ is typically overcome by enforcing sparsity via an entry-wise $\ell_1$ norm $\|A\|_{1,1} = \sum_{i,j=1}^n |A_{ij}|$ as a penalty term, like in eq. (4). While this covers the linear case, in section 4 we develop a technique to partially overcome problem (a) in the non-linear case, by reformulating $\|f\|_\epsilon$ as an entry-wise $\ell_1$ norm for a matrix derived from the parameters $\theta$ of the neural network $f_\theta$ approximating $f$.

**Remarks.** In the realm of neural ODEs one may be tempted to enforce sparsity in the neural network parameters $\theta$ directly. While this can be a sensible regularization scheme to improve generalization (Liebenwein et al., 2021), it does not directly translate into interpretable properties of the ODE $f_\theta$. For fully connected neural networks even a sparse $\theta$ typically leads to dense input-output connections such that $\|f\|_\epsilon = n^2$.

Alternatively, one may train a separate neural network $f_{i,\theta_i} : \mathbb{R} \times \mathbb{R}^n \to \mathbb{R}$ with parameters $\theta_i$ for each component $f_i$. Stacking the outputs of all $f_{i,\theta_i}$ we can train each network separately from the same training signal. For such a parallel setup we can enforce sparsity via $\|f_i\|_\epsilon^{\text{single}} := \sum_{j=1}^n \mathbf{1}\{\|\partial_j f_i\|_2 > \epsilon\}$ in the first layer parameters of each component neural network separately to "zero out" certain inputs entirely (Bellot et al., 2021). However, this differs from our sparsity measure $\|f\|_\epsilon$ in eq. (6). The latter allows large $\|f_i\|_\epsilon^{\text{single}}$ for *some* components $f_i$ *as long as the entire system is sparse*. The parallel approach did not perform better in empirical experiments, but is computationally more expensive.

# 4 Method

**Practical regularization.** Let us write out $f_\theta$ explicitly as a fully connected neural network with $L$ hidden layers parameterized by $\theta := (W^l, b^l)_{l=1}^{L+1}$, with $l$-th layer weights $W^l$ and biases $b^l$

$$f_\theta(Y) = W^{L+1}\sigma(\ldots \sigma(W^2\sigma(W^1 Y + b^1) + b^2)\ldots), \tag{7}$$

with element-wise non-linear activations $\sigma$. With this parameterization we now approximate our desired regularization $\|f_\theta\|_{\text{causal}}$ in terms of $\theta$. A major drawback of the natural candidate $\|f_\theta\|_\epsilon$ from eq. (6) is that it is piece-wise constant in $\theta$—an obstacle to gradient-based optimization. Instead, we aim at replacing $\|f\|_\epsilon$ with a differentiable surrogate. Ideally, we would like to use $\ell_1$ regularization on the strengths of all input to output connections $j \to i$ in $f_\theta$.

- **The linear case.** Recall that for $\mathcal{F}_{\text{lin}}$ it suffices to choose linear activation functions (specifically, $\sigma(x) = x$) and $b^l = 0$, such that $f_\theta(Y) = AY$ for some $A = W^{L+1} \cdot \ldots \cdot W^1$.[9] Hence, for $f_\theta \in \mathcal{F}_{\text{lin}}$, we can directly implement a continuous $\ell_1$ surrogate of the desired regularizer in terms of $\theta$

$$\|\theta\|_{\text{simple}}^{\text{lin}} := \|A\|_{1,1} = \|W^{L+1} \cdot \ldots \cdot W^1\|_{1,1}. \tag{8}$$

We then have $X_j \in pa_{f_\theta}(X_i)$ if and only if $A_{ij} \neq 0$. When restricting ourselves to $\mathcal{F}_{\text{lin}}$, using $\sigma(x) = x, b^l = 0$ together with the above sparsity constraint is thus a viable and theoretically sound method. In this case we infer $A$ directly from $\theta$, i.e., we identify $f$, from which the causal structure follows.

---

[8] $\|A\|_{\epsilon=0}$ and therefore also $\|A\|_{\text{causal}}$ is not a norm; it violates the triangle-inequality.

[9] When biases are non-zero, we are in the realm of inhomogeneous, linear, autonomous systems.

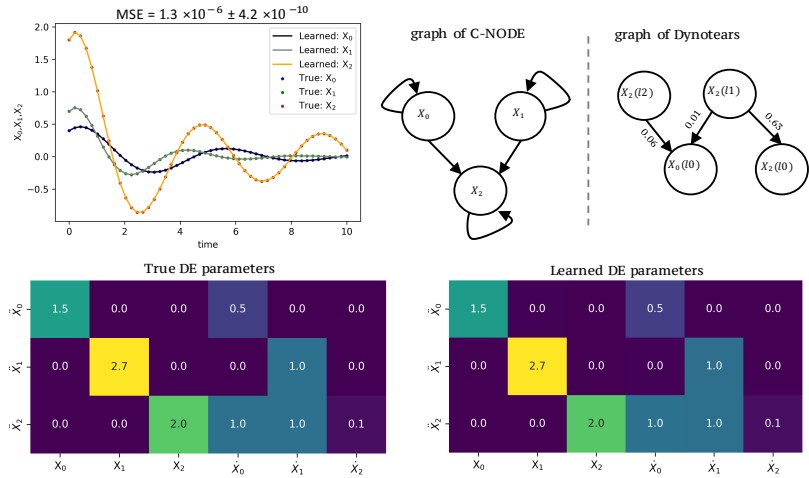

Figure 1: Exemplary second-order linear system.

- **The non-linear case.** In practice, we do not know whether $f^\star \in \mathcal{F}_{\text{lin}}$ a priori. Thus we remain open to the possibility of non-linear $f^\star$ via non-linear activation functions $\sigma$. Then $\|\theta\|_{\text{simple}}^{\text{lin}}$ is not equivalent to $\|f_\theta\|_{\text{causal}}$ anymore, because we may have $X_j \notin pa_{f_\theta}(X_i)$ despite $A_{ij} \neq 0$. In this case, we use the absolute weights for the regularizer:

$$\|\theta\|_{\text{simple}}^{\text{non-lin}} := \||W^{L+1}| \cdot \ldots \cdot |W^1|\|_{1,1} . \tag{9}$$

While we can still have $[|W^{L+1}| \cdot \ldots \cdot |W^1|]_{i,j} \neq 0$ even though $X_j \notin pa_{f_\theta}(X_i)$, $[|W^{L+1}| \cdot \ldots \cdot |W^1|]_{i,j} = 0$ always implies $X_j \notin pa_{f_\theta}(X_i)$ for $b^l = 0$. Hence, using $\|\theta\|_{\text{simple}}^{\text{non-lin}}$ as a regularizer aims at minimizing an upper bound of $\|f_\theta\|_{\text{causal}}$. We show empirically that enforcing this upper bound of the desired regularizer serves as an effective inductive bias to enforce sparsity in the causal connections.

From now on, we will drop the superscript of $\|\theta\|_{\text{simple}}$ when it is clear from context.

**Causal structure inference.** While we could read off the causal structure directly from $\theta$ via $A$ in the linear case, for non-linear $f_\theta$ we can validate our results via partial derivatives $\partial_j(f_\theta)_i$ over time, showing how each $\dot{X}_i$ depends on each $X_j$. Following the reasoning of the regularizer $\|f_\theta\|_\epsilon$ in eq. (6), we then reconstruct the causal relationships via $X_j \in pa_{f_\theta}(X_i)$ if and only if $\sum_{k=1}^{N} |\partial_j f_{\theta,i}(t_k)| > \epsilon$ for the $N$ observations at times $a = t_1 < \ldots < t_N = b$ and some threshold $\epsilon > 0$.[10] Thus we can still infer the causal structure in non-linear cases where evaluating whether our method identified the correct $f^\star$ is challenging (as we cannot compare parameters directly, but would have to compare a neural network to an analytically known function in symbolic form). The choice of $\epsilon$ is sensitive to the scale of the data, which we account for by normalizing data before training. Empirically we did not observe strong dependence of the inferred causal structure on the choice of $\epsilon$ for normalized data. A simpler method is available to determine the *absence* of causal dependencies in the non-linear setting: if the entry $[|W^{L+1}| \cdot \ldots \cdot |W^1|]_{i,j}$ is (close to) zero, then $X_j \notin pa_{f_\theta}(X_i)$.

**Summary.** C-NODE adds a practical, differentiable regularizer $\lambda \|\theta\|_{\text{simple}}$ with a tuneable regularization parameter $\lambda$ to the NODE loss to recover sparse dynamics corresponding to our regularized goal. Our regularizer captures $\|f\|_{\text{causal}}$ perfectly in the linear case and enforces minimization of an upper bound in the non-linear case. We also devised a method to recover the causal structure from linear and non-linear $f_\theta$. In Section 5 we show empirically that potentially remaining theoretical unidentifiability does not practically impede C-NODE from recovering $f^\star$, allowing for accurate predictions under variable and system interventions. In Appendix D we discuss extensions for latent dynamics and data from heterogeneous environments.

## 5   Experiments

We now illustrate the robustness of our method in several case studies. The general principles readily extend to more complex model classes. Among several methods developed for causal inference from time-series data

---

[10]This is but one simple approach to estimate practically whether $\|\partial_j f_{\theta,i}\|_2 \neq 0$, i.e., whether the $i$-th output of the neural network $f_\theta$ depends on the $j$-th input. While a host of more sophisticated methods may be used here, we found this simple approach sufficient in our experiments.

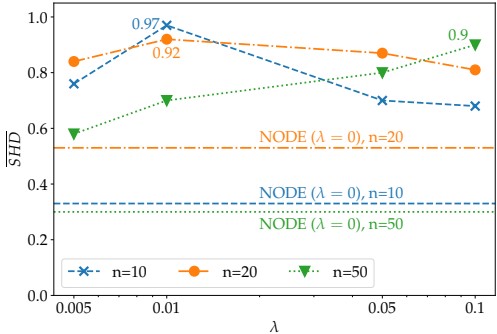

Figure 2: Dependence on the sparsity parameter $\lambda$. Dashed lines labeled $\lambda = 0$ refer to vanilla NODE.

Table 1: Experimental results using synthetic datasets with $n \in \{10, 20, 50\}$, varying noise level $\sigma$, and sampling irregularity (irr). SHD is the structural Hamming distance and $\overline{\text{SHD}} = 1$ - SHD.

| $\sigma$ | dim | irr = 0.0 | | | irr = 0.2 | | | irr = 0.5 | | | irr = 0.7 | | |
|---|---|---|---|---|---|---|---|---|---|---|---|---|---|
| | | $\overline{\text{SHD}}$ | TPR | TNR | $\overline{\text{SHD}}$ | TPR | TNR | $\overline{\text{SHD}}$ | TPR | TNR | $\overline{\text{SHD}}$ | TPR | TNR |
| | 10 | 0.97 | 0.95 | 0.98 | 0.98 | 0.97 | 0.98 | 0.93 | 0.97 | 0.91 | 0.92 | 0.93 | 0.91 |
| 0.0 | 20 | 0.92 | 0.82 | 0.97 | 0.84 | 0.72 | 0.88 | 0.85 | 0.74 | 0.88 | 0.86 | 0.75 | 0.89 |
| | 50 | 0.90 | 0.71 | 0.92 | 0.85 | 0.67 | 0.87 | 0.92 | 0.69 | 0.96 | 0.93 | 0.70 | 0.96 |
| | 10 | 0.88 | 0.81 | 0.92 | 0.87 | 0.80 | 0.92 | 0.86 | 0.84 | 0.88 | 0.67 | 0.48 | 0.87 |
| 0.05 | 20 | 0.86 | 0.78 | 0.89 | 0.84 | 0.72 | 0.88 | 0.72 | 0.59 | 0.75 | 0.69 | 0.53 | 0.73 |
| | 50 | 0.91 | 0.64 | 0.94 | 0.90 | 0.69 | 0.93 | 0.90 | 0.67 | 0.93 | 0.89 | 0.65 | 0.93 |
| | 10 | 0.78 | 0.68 | 0.74 | 0.71 | 0.68 | 0.71 | 0.65 | 0.77 | 0.58 | 0.50 | 0.60 | 0.51 |
| 0.1 | 20 | 0.82 | 0.79 | 0.82 | 0.79 | 0.74 | 0.80 | 0.68 | 0.53 | 0.74 | 0.61 | 0.48 | 0.65 |
| | 50 | 0.86 | 0.65 | 0.90 | 0.89 | 0.59 | 0.93 | 0.87 | 0.61 | 0.91 | 0.86 | 0.68 | 0.89 |

based on Granger causality (Tank et al., 2021; Hyvärinen et al., 2010; Runge et al., 2019; Amornbunchornvej et al., 2019), we compare to `Dynotears` (Pamfil et al., 2020), because it outperforms most competing methods in their evaluation. For system identification, we also compare to GroupLasso (Bellot et al., 2021) and PySINDy (Brunton et al., 2016; de Silva et al., 2020). Details on all parameter settings, evaluation, and implementation choices are provided in Appendix C.

**Linear ODEs.** We first study second-order, homogeneous, autonomous, linear ODEs

$$\ddot{X} = W_1 \dot{X} + W_2 X. \tag{10}$$

We begin with $n = 3$ and randomly chosen true weight matrices $W_1^\star, W_2^\star$ from which we generate $X^\star$ using a standard ODE solver, see Appendix C. Figure 1 shows that our method not only accurately predicts $X^\star$, but it also identifies $W_1^\star, W_2^\star$ within a maximum absolute difference of 0.018. Thus the causal graph is also inferred correctly. The poor performance of `Dynotears` may be due to cyclic dependencies in $W_1^\star, W_2^\star$.

We extend these results to study the *scalability* of our method and its performance when the observations are *irregularly* sampled (a fixed fraction of observations is dropped uniformly at random) with *measurement noise* (additive zero-mean Gaussian noise with standard deviation $\sigma$). The data generation specifics for three synthetic datasets with 10, 20, and 50 variables are described in Appendix C. We evaluate the inferred causal graph using the structural hamming distance (SHD) (Lachapelle et al., 2019) for the fraction of wrongly predicted edges, the true positive rate (TPR), and the true negative rate (TNR). The results in Table 1 show that C-NODE performs well for non-noisy data ($\sigma = 0$) and is robust to randomly removing samples from the observation. Accuracy drops with increasing noise levels, which is further exacerbated by sampling irregularities, suggesting improved robustness to observation noise as an interesting direction for future work. In Figure 2 we show the dependence of $\overline{\text{SHD}} = 1 - \text{SHD}$ on the regularization parameter $\lambda$ for high-dimensional systems following eq. (10). C-NODE clearly outperforms vanilla NODE for causal structure inference with only moderate sensitivity to $\lambda$ even though they achieve similar predictive accuracy. This indicates that for sparse systems unidentifiability may indeed be a problem (as hypothesized in section 3) in that the solution trajectory can be perfectly reconstructed also by "false" dense systems. This provides evidence that regularization is indeed crucial for recovering the correct sparse ground truth dynamics. In

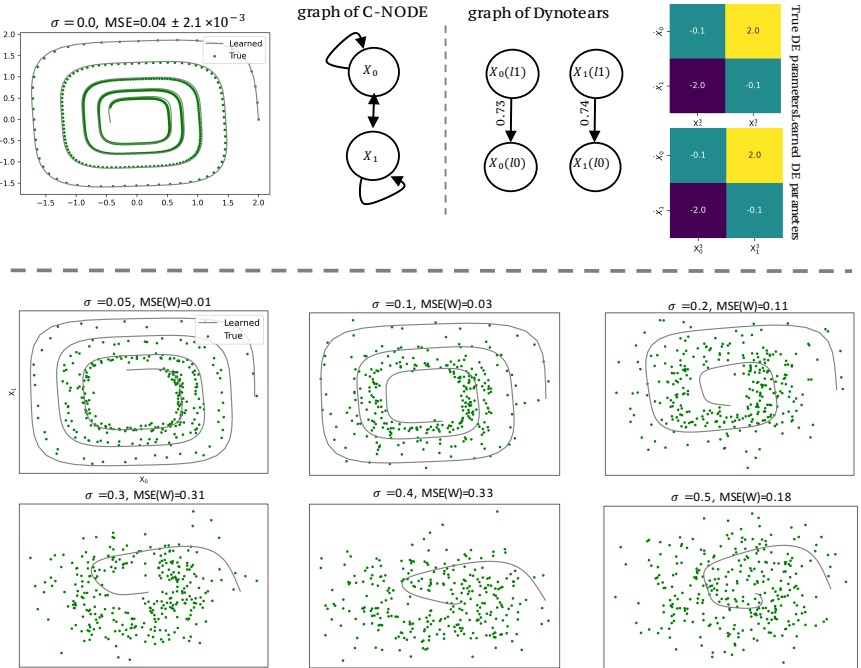

Figure 3: Results for the spiral ODE.

Figures 8 and 9, we provide further results and show that C-NODE also outperforms both vanilla NODE Chen et al. (2018), GroupLasso (Bellot et al., 2021) and PySINDy (de Silva et al., 2020).

A concrete application example for a common (synthetic) chemical reaction network of transcriptional gene dynamics is also provided in Appendix E.2.

**Spiral ODEs.**  The spiral ODE model is given by

$$\dot{X}_0 = -\alpha X_0{}^3 + \beta X_1{}^3, \qquad \dot{X}_1 = -\beta X_0{}^3 + \alpha X_1{}^3 \tag{11}$$

and features cyclic dependencies and self-loops. We follow the parameterization in Chen et al. (2018). While `Dynotears` fails to estimate the cyclic causal graph, Figure 3 shows that C-NODE infers the actual ODE parameters $\alpha, \beta$ and thus the causal structure correctly. Again, Figure 3 illustrates that predictive performance slowly degrades as observation noise levels increase raising the mean-squared error (MSE) of the inferred adjacency matrix substantially for higher variance noise. However, the deduced causal structure remains correct.

**Lotka-Volterra ODEs.**  The Lotka-Volterra predator-prey model is given by the non-linear system

$$\dot{X}_0 = -\alpha X_0 - \beta X_0 X_1, \quad \dot{X}_1 = -\delta X_1 + \gamma X_0 X_1. \tag{12}$$

We use the same parameters as Dandekar et al. (2020) and ReLU activations for non-linearity. Figure 4 shows the excellent predictive performance of C-NODE (left). Because of the non-linearity, we resort to our non-linear causal structure inference method and show the partial derivatives $\partial_j f_{\theta,i}$ for the learned $f_\theta$ in Figure 4 (right). For example, from eq. (12) we know that $\partial \dot{X}_0 / \partial X_1 = -\beta X_1$ and indeed $\partial_1 f_{\theta,0}$ in Figure 4 (right) resembles $X_1$ in Figure 4 (left) up to rescaling and a constant offset. Similarly, the remaining dependencies estimated from $f_\theta$ strongly correlate with the true dependencies encoded in eq. (12), giving us confidence that $f_\theta$ has indeed correctly identified $f^\star$.

**Interventions.**  To back up this claim, we assess whether we can predict the behavior of systems under interventions. We consider C-NODEs trained on observational data (without interventions) from Figures 1 (simple linear system) and 4 (Lotka-Volterra). We apply two types of interventions: (1) a system intervention replacing one entry of $A^\star$ via $\tilde{A}_{00} := 8A^\star_{00}$ (for example, a temperature change that increases some reaction rate eightfold) in the linear setting, and (2) variable interventions $X_0 := 0.4$ in the linear as well as $X_0 := 1$

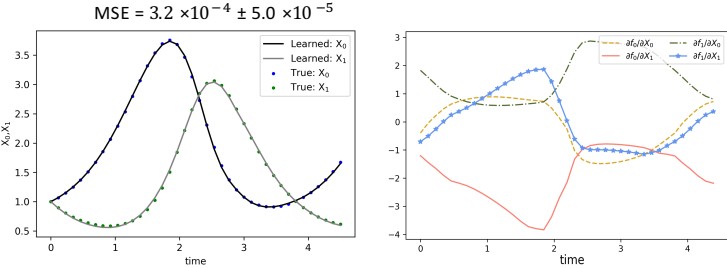

Figure 4: Results for the Lotka-Volterra example.

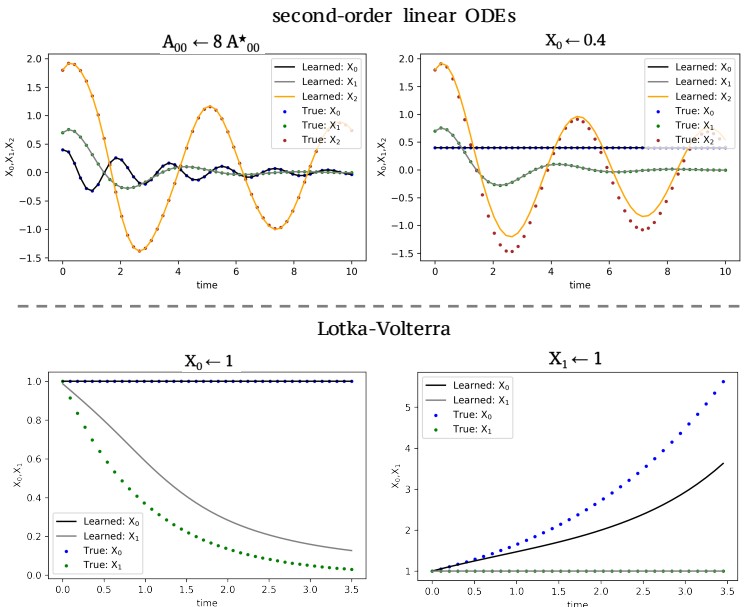

Figure 5: Predictions under interventions.

and $X_1 := 1$ in the non-linear Lotka-Volterra setting (for example, keeping the number of predators fixed via culling quotas and reintroduction). Figure 5 (top left) shows that C-NODE successfully predicts the linear system's evolution under the system intervention. For the variable intervention in the linear setting (top right), $X_1$ correctly remains unaffected, while the new behavior of $X_2$ is predicted accurately.

In the Lotka-Volterra example, both variable interventions impact the other variable. Fixing either the predator or prey population should lead to an exponential increase or decay of the other, depending on whether the fixed levels can support higher reproduction than mortality. Figure 5 (bottom row) shows that our method correctly predicts an exponential decay (increase) of $X_0$ ($X_1$) for fixed $X_1 := 1$ ($X_0 := 1$) respectively. The quantitative differences between predicted and true values stem from small inaccuracies in the predicted parameters which amplify exponentially to seemingly large quantitative differences.

**Real single-cell RNA-seq data.** Finally, we apply C-NODE to learn gene-gene interactions. Gene (feature) interactions, also known as causal dependencies between genes, are often represented as a gene regulatory network (GRN) where nodes correspond to genes and directed edges indicate regulatory (or causal) interactions between genes. GRN inference from observations is known to be an exceptionally difficult task Perkel (2022). The inferred GRN is expected to be sparse as regulatory genes known as transcription factors do not individually target all genes. Therefore, sparsity in the number of interactions is essential.

In this experiment, we first explore how pruning improves GRN inference from human hematopoiesis single-cell multiomics data Luecken et al. (2021) (GEO accession code: GSE194122). We select a branch of data in which hematopoiesis stem cells (HSCs) differentiate into Erythroid cells with a total of 280 cells (or samples). The count matrix is normalized to one million counts per cell. Figure 6 (top row) shows a UMAP representation (McInnes et al., 2018) of the data where each point corresponds to a cell colored by cell type (left) and an

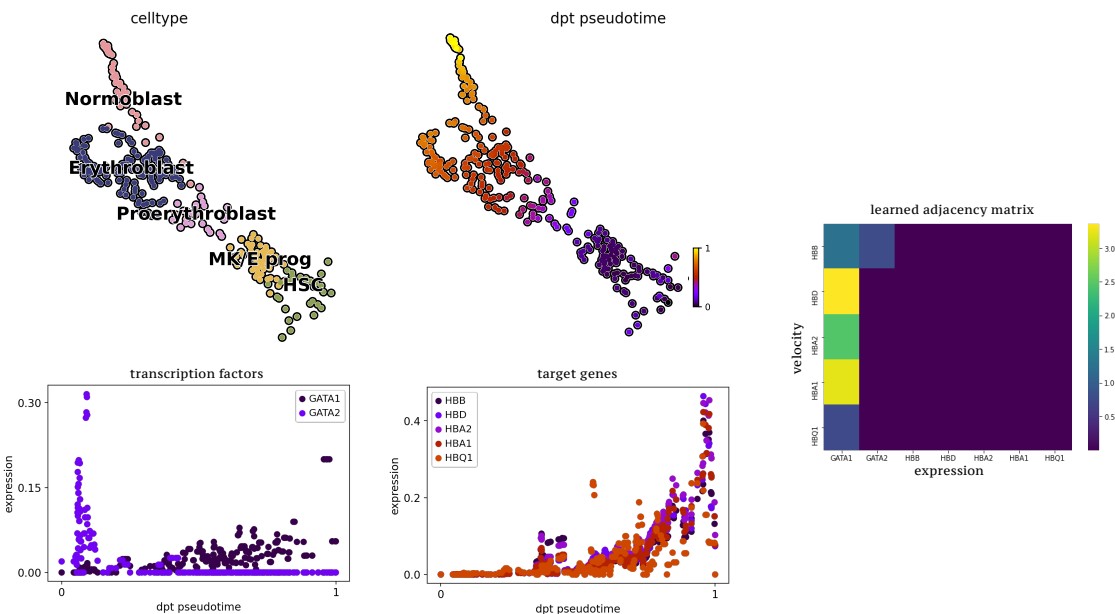

Figure 6: Gene regulatory network inference results using human bone marrow data with 7 genes.

inferred continuous pseudotime (right). This pseudotime aims at identifying how far a cell has advanced in the differentiation process and is inferred via a diffusion map based manifold learning technique called *dpt* (Haghverdi et al., 2016) on 2,000 highly variable genes (Wolf et al., 2018; Bergen et al., 2020). We take measured gene expression levels over pseudotime $t$ as our observations $X^\star(t)$.

In this setting, domain knowledge asserts that GATA genes (*GATA1*, *GATA2*) regulate the expression of hemoglobin subunits (*HBB*, *HBA1*, *HBA2*, *HBD*, *HBQ1*) (Ding et al., 2010; Johnson et al., 2002; Suzuki et al., 2013; Shearstone et al., 2016). The normalized expression of genes related to these subunits over pseudotime is presented in Figure 6 (middle row). The expressions are scaled between 0 and 1 for each gene before training. We first apply C-NODE to these 7 genes with known ground truth. The bottom row of Figure 6 shows the row-wise normalized absolute values of the adjacency matrix inferred by C-NODE. Our approach properly assigns hemoglobin subunit changes to GATA genes, even though visually the hemoglobin target genes appear to be more correlated among themselves than with the GATA drivers.

We expect the regulatory elements and their target genes to be similar across species. To show that the previous results are stable across species, we next apply C-NODE to mouse single-cell RNA-seq data from (Pijuan-Sala et al., 2019) (GEO accession number: GSE87038) where blood progenitors similarly differentiate into Erythroid cells. Consistent with previous results, we also observe in Figure 11 that hemoglobin genes depend on GATA genes in Erythroid lineage (more details are discussed in Appendix E.3).

Finally to study the scalability of the proposed method and the effectiveness of the sparsity regularizer, we select 529 highly-variable genes from the whole human hematopoiesis data Luecken et al. (2021) and apply C-NODE on cells from the Erythroid lineage. Many of those genes are spurious for the Erythroid lineage as their expression does not change along the lineage. After training, the model selects 141 key genes as important with at least one interaction to other genes. Our first observation is that the important genes are ranked as highly variable only for the Erythroid lineage (Figure 14) which shows that C-NODE avoids using spurious features for prediction (Figure 7). Using the chromatin accessibility features available in the human immune cells datasets as reference, we also observe 22 regulatory genes known as transcription factors among those important genes. Assessed by the literature, all the 22 transcription factors are important for the differentiation of the Erythroid lineage. The expression and the prediction of four transcription factors are shown in Figure 7, left. Other transcription factors are listed in Table 2.

Since ground truth dynamics are not known for GRNs, validating the inferred interactions is challenging. In order to assess the biological relevance of the learned interactions, we perform Gene Set Enrichment Analysis (GSEA) via the enrichr method Chen et al. (2013) (discussed in Appendix E.3). The enrichment test also captures many relevant processes including hematopoietic stem cell differentiation, Erythrocyte differentiation,

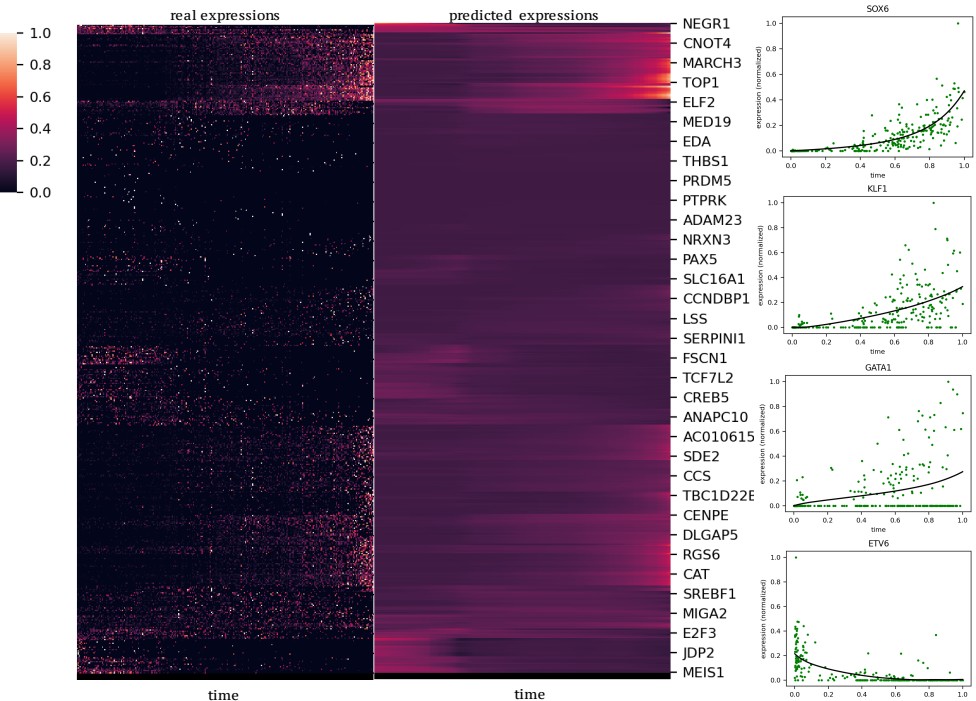

Figure 7: Prediction performance using human bone marrow data with 529 genes. Shown are the predictions for 141 important features (on left), and selected transcription factors (on right).

and some immune related signaling pathways. This provides evidence that indeed key regulatory processes for the differentiation have been captured by our model.

## 6 Conclusion

We proposed C-NODE, an approach to identification and causal structure learning of (sparse) ODE-based dynamical systems using Neural ODEs. First, we observe that even if a method achieves perfect predictive accuracy, it may not be able to predict the system's behavior under interventions, as ODE identification is generally ill-posed. Therefore, a key focus of our work lies on the predominantly neglected issue of restricting the search space in meaningful ways to recover the underlying sparse system and causal structure.

We devised a simple and practical method to extract causal dependencies (including cyclic relationships) from a learned neural ODE derivative network. We then demonstrated that C-NODE performs well on causal structure identification for a wide variety of settings and further corroborated our findings by correctly predicting the effect of different forms of interventions targeting both the evolving variables as well as parameters of the governing ODE itself.

In our experiments we analyze C-NODE on synthetic and real-world gene regulatory data with varying numbers of variables, noise levels, and irregular sampling intervals. While unidentifiability indeed affects vanilla NODE, C-NODE still reliably infers sparse causal structures. Going forward, our results suggest an in-depth analysis of the conditions under which unidentifiability manifests itself in practice as a fruitful direction for future work. At the same time, extending C-NODE for successful hypotheses generation in high-dimensional real datasets with stochasticity, delay, unobserved confounding, or heterogeneous environments is an exciting challenge for further research.

Given these limitations, we highlight that caution must be taken when informing consequential decisions, e.g., in healthcare, based on causal structures learned purely from observational data. At the same time, we hope that causal modelling of dynamical systems broadly and C-NODE in particular can be valuable tools for hypothesis-generation in various scientific domains to suggest promising experiments for in-depth follow up.

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
