# OpenReview forum: "Beyond Predictions in Neural ODEs: Identification and Interventions"
_TMLR — Rejected by TMLR_

### Review · Reviewer_CgRJ · 2022-10-24

**Summary Of Contributions:**

This work addresses the problem of estimating the vector field underlying an ordinary differential equation (ODE) from a single observed trajectory, where all the variables in the system are assumed to be observed.
The authors models the ODE using a neural network (following the neural ordinary differential equation framework).

As a major contribution, the authors proposes a regulariser for the neural network that encourages sparsity  (in a sense that a variable is influenced by a few other variables) to address the ill-posed-ness of the estimation problem.
The authors define a causal relation for a given vector field, and propose a method to infer causal relations among variables based on the definition.





**Audience:**

No

**Claims And Evidence:**

No

**Requested Changes:**

Please address the above issues.

**Strengths And Weaknesses:**

## Strengths
The problem addressed in this work is of practical interest. In particular, the focus on the prediction ability under unseen interventions is interesting. The paper is generally well-written and easy to follow.


## Weaknesses

The major weakness is that the utility of the proposed regulariser is not convincingly established, as detailed in the following:

1. While this paper is an empirical work, the argument for the proposed method lacks a theoretical basis. Specifically, I am not sure about the claim made on Page 7 in the non-linear case. Why does the function $||\cdot||^{\mathrm{non-lin}}_{\mathrm{simple}}$ in Eq. (9) upper bound the desired regulariser? This claim does not have any proof. Given that the function (the element-wise $\ell^1$-norm of the matrix product) only regularises the network parameter, it is not so easy to see why the stated claim holds. Can this be empirically shown?

2. A related issue is in the proposed inference method for causal structures (Page 7). In this paragraph, the authors propose to check the partial derivative of each component function as $\sum_{k=1}^N |\partial_j f_{\theta, i}(t_k)|>\epsilon$ for times $t_k$.
Assuming that what is meant by $|\cdot|$ is the $L^2$ norm, this check seems to have the same problem as mentioned on the previous page; see the paragraph containing "$L^2$-norms of partial derivatives are difficult to evaluate efficiently and accurately.". This requires some explanation.
3. I find it difficult to see the benefit of the proposed regulariser from experiments, particularly for non-linear systems.
   * The effect of having a regularise is only investigated for linear ODEs. Regarding the identification success results reported in the following experiments (spiral/Lotka-Volterra ODEs), it is not clear if these results are due to the use of NODEs or the proposed regulariser.
   * Relatedly, the authors use relatively low-dimensional non-linear systems (except for the final experiment Real single-cell RNA-seq data).  Having a synthetic higher-dimensional toy model would make convincing the claim that NODEs scale in dimensionality.


### Minor comments
* The proposed norm is not everywhere differentiable
* Page 5: $k$ non-zero enties. The variable $k$ is not defined?
* Page 7: Requiring the partial derivative to be in $L^2$ seems to be restrictive (e.g., excluding polynomials as in the spiral ODEs).
* Page 9: What is $A^*$?

---

### Review · Reviewer_68WX · 2022-11-09

**Summary Of Contributions:**

The paper introduces a simple and differentiable way to encourage sparsity in a feed forward neural network that would ensure independence of certain outputs from certain inputs.  With this sparsity regularizer, the network is used and trained as the derivative function $f$ of a NODE solver.  Since non-zero values in the regularizer encode direct dependence of the variables, the submission denotes their regularized version C-NODE, for causal. Empirical demonstrations include identifiability of the causal connections under various conditions, demonstration of the model fit, interventional tests, and an application to gene interaction data.


**Audience:**

Yes

**Broader Impact Concerns:**

no concerns

**Claims And Evidence:**

No

**Requested Changes:**

### Assumptions
The manuscript assumes, similarly to the Granger Causality, that the data comes from a causally sufficient system. However, no mention of it anywhere. What would happen if a variable directly affecting multiple other variables was not included. Please, either explicitly state that the approach is unable to handle cases with latent confounders or provide rationale why it would together with additional experiments to demonstrate that it does work.
### Definitions
- Please, to avoid ambiguities, explicitly define SHD where it is introduced.
### Experiment description
- Please indicate ground truth graph density as, for example, number of edges over the total possible edges $n^2$, or any other appropriate but clear measure of density.
- Please describe how the graphs were generated for experiments in Figure 2 and Table 1. Were they totally random or had some interesting structure, like having strong connectivity to mimic biological modules? If random, then what procedure was used to generate the randomness.
- Please report the value of $\lambda$ used for experiments in Table 1, right in the caption of the table or somewhere else nearby.
- Figure 2 and Table 1 provide a single point estimate for each parameter setting. Does that mean there was only a single experiment conducted for each setting? Was it done for a random graph or a few graphs and then the best(?) or the worst(?) result reported?
### Experiments
1. Please, run experiments on a range of random graphs (e.g. 100) and report results with error bars. Better yet to show a box-plot or similarly informative view for Figure 2. Table 1, probably would do with error-bars/std.
2. Please, also provide experiments demonstrating the degree of stability of the model to changes in initialization for experiments in Table 1.
3. Please explain how the density of the 10, 20, and 50 node graphs was defined. It is intriguing why in the larger graphs C-NODE tends to underestimate the number of edges. It looks from TPR that it produces a result sparser than the ground truth, as more visible for 50-node graphs than small. I suspect the smaller graphs are relatively denser than the larger due to some inbalanced way of setting sparsity. Please, clarify.
4. Would it be possible to demonstrate the model fit on a larger graph than the 2-node graphs used?
### Figures
- All figures would benefit with expanded captions that contain an explanation of the figure's content.
- Figure 2 would potentially be clearer if a standard subfigure format is used with explanations.
- It is unclear in Figure 1 and 2 whether the Learned DE parameters are learned by Dynotears of C-NODE. Figure 2 is especially incomprehensible in this regard.
- Similarly, it is unclear which method produced the figures with the spiral fits. They are not clearly marked.
### Notation
On page 3 the notation is hard to follow. A good notation is hard to come up with but still, please try to correct things like a sequence

**Strengths And Weaknesses:**

### Strengths
1. The paper is generally well-written. The introductory sections up to Section 4 especially so.
2. The idea of the regularizer is simple and elegant.
3. The experiments target appropriate questions
### Weaknesses
1. Experiments are poorly explained, executed, or performed on small systems.
2. The technical details are not well delivered, which includes some definitions, details of experiments, and notation.
3. Figures need work to be useful for a reader.

---

### Review · Reviewer_UPjD · 2022-11-22

**Summary Of Contributions:**

Uncovering underlying dynamics of a system from observational data can hugely benefit area of discovering new knowledge. In the general case, unless the observational data covers the complete data-distribution, it is not possible to uncover the true dynamics without further interaction with the system. However, one can still hope to come up with inductive biases that can enable us to discover the true dynamics on certain data-distributions.

The contribution in this paper is along similar lines---the authors propose a regularization scheme for neural ODEs that can recover the underlying dynamics of the system on some benchmarks. The central idea behind the proposed regularizer is to limit the number of inputs that will impact the output. The authors use a heuristic to measure the impact in the non-linear neural network case.

Finally, the authors evaluate their method on a number of simulatoed benchmarks.

**Audience:**

Yes

**Claims And Evidence:**

Yes

**Requested Changes:**


1. The figures should be more self-contained. The caption should summarize what the figure is, what the reader should look at when looking at the figure, and what conclusions the authors are drawing from the figures. Doing so will improve readibility, since the reader will not have to jump between the text and the figure multiple times to understand the figure. (It's okay to repeat information in a caption that is already written in the text).


2. How are the weight matrics W_1^* and W_2^* chosen for the linear ODEs experiment? The text says randomly, but there are many ways to sample these weights randomly. Looking at Figure 8, I can see that the ground truth causal dependency is sparse; are W_1^* and W_2^* chosen to make the ground truth sparse? I would imagine the dependency to not be sparse if W_1^* and W_2^* were sampled uniformly random to be in a range.

3. If W_1^* and W_2^* were chosen to make the ground truth sparse, then it would be a good idea to add results for an experiment when they were not chosen to make the ground truth interactions to be sparse. If the ground truth dependency was dense, I would expect that C-Node would perform worse than NODE for large values of lambda. Having such an experiment would clarify when we can expect C-NODE to perform better than NODE. The current write-up might falsely suggest that ground truth depedencies are always sprase, and that C-NODE is a strict improvement over NODE for all values of lambda.

4. Instead of saying "We follow the parameterization in Chen et al. (2018)," it would be more clear to mention the parameterization in the paper since the detail of the parameterization is an important part of the experiment. If there are too many details, the author should at-least summarize the characerstics of the parameterization proposed by Chen et. al. (2018).

5. Similarly, the authors should either mention, or at-least summarize the parameters used by Dandekar et al. (2020).

6. Maybe I'm misunderstanding something, but did the authors run node on the same RNA-seq data? How does C-NODE perform compared to NODE?

7. The heuristic proposed for the non-linear network regularization in equation 9 can be explained more clearly.

8. It seems like equation 9 is too strong of a constraint to measure if X_i impacts X_j. Even if all out-going connections from X_i are set to be zero, in which case X_i does not impact any output, the ith,jth entry in the product of absolute values of weights can still be non-zero. In-fact, to make the ith/jth entry zero, the network would have to become far more sparse than needed to just mask the impact of X_i. Isn't the proposed metric too strong of a regularizer as a result?

9. Using partial derivative to determine causal relation: Doesn't the partial derivative only tells if changing the input slightly will impact the output? It is possible for the output to depend on an input, and yet the partial derivative to be zero. Additionally, the value of X_j matters. For example, if X_j = 0, the partial derivative can still be very large when X_j plays no role in the prediction. A better hueristic would be |\partial_j f_{\theta,i}  X_j| as opposed to |\partial_j f_{\theta,i}| i.e. gradient times the input value tells us how much the output would change if we removed the input under the assumption that the network is linear. Gradient alone, on the other hand, doesn't have a similar interpetation, and the gradient of a prediction w.r.t an input can be very large even when the input is zero.

### Other comments that don't necessarily have to be addressed:

"The hope is that machine learning may sometimes be able to deduce true laws of nature purely from observational data, promising reliable predictions not only within the observed setting, but also under interventions"

Isn't it trivially true that this hope is meaningless? The only way to verify if a system has done this is to verify under interventions, which requires interaction. No matter what algorithm one comes up with, there would be no way to know if it has learned the true dynamics from observational data without collecting more data.


"We discuss potential regularizers to enforce sparsity in the number of causal interactions such that variables depend on few other variables, a common assumption in modern causal modeling"

There is little evidence to support that sparsity is a good assumption when learning from sensory data. I know that this assumption is widely accepted in the field of causal modeling, but it has never been motivated well. However, because this is a commonly accepted belief, I do not expect the authors to defend it and will review the paper as if the assumption is valid.


**Strengths And Weaknesses:**

The paper is clearly written, and does a thorough evaluation of the proposed method. Overall, it's a good paper that can be accepted after some changes. Please see the next section for most of my comments.

---

### Decision · Action_Editors · 2023-02-28

**Recommendation:** Reject

**Comment:**

See my comments above.

**Audience:**

The authors did not provide a rebuttal for this paper. The paper is on the subject of causality for time series data, which indeed attracts the interest of some individuals in the TMRL audience.

**Claims And Evidence:**

The authors did not provide a rebuttal for this paper. The reviewers have made specific comments regarding the limited experimental evaluation and the lack of proper acknowledgement of the implicit assumptions made by the method.